# Selective PROTAC-mediated degradation of SMARCA2 is efficacious in SMARCA4 mutant cancers

Jennifer Cantley[1], Xiaofen Ye[2], Emma Rousseau[1], Tom Januario[2], Brian D. Hamman[3], Christopher M. Rose [2], Tommy K. Cheung[2], Trent Hinkle[2], Leofal Soto[1], Connor Quinn[1], Alicia Harbin[1], Elizabeth Bortolon[1], Xin Chen[1], Roy Haskell[1], Eva Lin[2], Shang-Fan Yu[2], Geoff Del Rosario[2], Emily Chan[2], Debra Dunlap[2], Hartmut Koeppen[2], Scott Martin[2], Mark Merchant[2], Matt Grimmer[2], Fabio Broccatelli[2], Jing Wang[1], Jennifer Pizzano[1], Peter S. Dragovich[2], Michael Berlin[1] & Robert L. Yauch [2] ✉

The mammalian SWItch/Sucrose Non-Fermentable (SWI/SNF) helicase SMARCA4 is frequently mutated in cancer and inactivation results in a cellular dependence on its paralog, SMARCA2, thus making SMARCA2 an attractive synthetic lethal target. However, published data indicates that achieving a high degree of selective SMARCA2 inhibition is likely essential to afford an acceptable therapeutic index, and realizing this objective is challenging due to the homology with the SMARCA4 paralog. Herein we report the discovery of a potent and selective SMARCA2 proteolysis-targeting chimera molecule (PROTAC), A947. Selective SMARCA2 degradation is achieved in the absence of selective SMARCA2/4 PROTAC binding and translates to potent in vitro growth inhibition and in vivo efficacy in SMARCA4 mutant models, compared to wild type models. Global ubiquitin mapping and proteome profiling reveal no unexpected off-target degradation related to A947 treatment. Our study thus highlights the ability to transform a non-selective SMARCA2/4-binding ligand into a selective and efficacious in vivo SMARCA2-targeting PROTAC, and thereby provides a potential new therapeutic opportunity for patients whose tumors contain SMARCA4 mutations.

The multi-subunit switch/sucrose non-fermentable (SWI/SNF or BAF) complex facilitates the remodeling of chromatin to regulate key cellular processes including transcriptional regulation and DNA repair[1,2]. Catalytic function is conferred by one of two mutually exclusive ATP-dependent helicases, SMARCA2 and SMARCA4, which share strong protein sequence homology[3]. In addition to a highly conserved ATPase domain (93% identity), both proteins contain a conserved bromodomain (96% identity) (BD) that can interact with acetylated chromatin. SWI/SNF has gained a great deal of attention in cancer biology, as

tumor sequencing studies have revealed that ~20% of human cancers harbor mutations in specific core or accessory components of the complex[4,5]. In particular, loss of function mutations in *SMARCA4* are enriched in subsets of multiple malignancies, with the highest prevalence of homozygous mutations occurring in non-small cell lung cancer (NSCLC)[6,7]. Aberrant chromatin remodeling caused by SMARCA4 mutations can result in the disruption of enhancer accessibility and accumulation of Polycomb repressive complexes 1 and 2 across the genome[8,9].

[1]Arvinas, LLC, 5 Science Park, New Haven, CT 06511, USA. [2]Genentech, 1 DNA Way, South San Francisco 94080, USA. [3]HotSpot Therapeutics, Inc. 1 Deerpark Dr., Ste C, Monmouth Junction, NJ 08852, USA. ✉e-mail: bobyauch@gene.com

Strategies to therapeutically target SMARCA4 mutant (SMARCA4$^{mut}$) cancers have focused on the identification of vulnerabilities that may be conferred in the context of the mutant state[10]. Most notably, functional genomic screens to identify gene dependencies have identified the paralog, SMARCA2, as a synthetic lethality in cancers with inactivated SMARCA4[11,12]. Although the ATPase activity of SMARCA2 is required for the proliferation of SMARCA4$^{mut}$ cells, bromodomain function is dispensable, as highlighted by the failure of inhibitors to the SMARCA2/4 bromodomain to suppress cell growth[13]. Such findings have subsequently led to efforts to discover ATPase inhibitors, however the currently described inhibitors are dual inhibitors of SMARCA2/4 and are hampered by dose-limiting tolerability issues, preventing the full in vivo exploration of anti-tumor activity[14]. Preclinical genetic studies would indicate that achieving selective inhibition of SMARCA2 will likely be essential for a successful therapeutic. Whereas the germline knockout of *Smarca2* produced viable mice that are slightly larger than control littermates but show no other overt phenotypes, the knockout of *Smarca4* is embryonic lethal and conditional deletion of *Smarca4* has been associated with multiple tissue-specific phenotypes[15–19]. More importantly, the co-deletion of *Smarca2* and *Smarca4* in adult mice was lethal due to vascular defects[20]. Hence, SMARCA2 inhibitors with improved selectively over SMARCA4 will likely be required to achieve safe and maximal inhibition of SMARCA2 in this context.

Proteolysis targeting chimeras (PROTACs) represent an emerging therapeutic modality to induce the degradation of target proteins by recruiting the protein of interest to an E3 ubiquitin ligase, leading to the subsequent tagging of the protein for proteasome-mediated destruction through the addition of ubiquitin[21,22]. PROTACs offer several advantages over classical small molecule inhibitors, as they circumvent the requirement to employ ligands targeting the enzymatic function of the given target protein and they can function in a substoichiometric manner enabling sustained pharmacodynamic effects at lower systemic exposures. Importantly, selective degradation using warhead ligands with nonselective binding properties has been demonstrated with PROTACs[23,24]. Although the mechanisms underlying selective degradation remain to be fully elucidated, the ability to form protein-protein interactions between the target protein and E3 ligase within the ternary complex can contribute to a more productive and selective degrader[23,25]. Hence, warhead selection and the choice of E3 ligase will play a critical role in determining whether selective degradation could be achieved using a PROTAC with equivalent binding affinities to multiple substrates.

In this work we use otherwise inert ligands with equivalent binding affinities to the bromodomains of SMARCA2/4 and the 5th bromodomain of PBRM1 to develop a VHL-based PROTAC exhibiting potent and moderately selective degradation of SMARCA2. The VHL-SMARCA2 PROTAC elicits enhanced growth inhibitory effects both in vitro and in vivo in SMARCA4$^{mut}$ cancer models relative to SMARCA4 wild-type (SMARCA4$^{wt}$) models, in the absence of considerable tolerability issues. In contrast to a previously described SMARCA2/4 ATPase inhibitor[14] and PROTAC[26], these findings provide the an example of a selective SMARCA2 targeting agent and provide pharmacologic support of this previously defined synthetic lethality in SMARCA4$^{mut}$ cancers.

## Results

### Identification of SMARCA2-selective PROTAC, A947
To identify potent and selective PROTACs targeting SMARCA2, we linked a small-molecule ligand capable of binding the bromodomains of SMARCA2/4 and PBRM1 (5th BD) to a VHL-targeting moiety (information regarding chemical synthesis can be found in the Methods and as a Supplementary Note in the Supplemental Information). This work led to the identification of PROTAC, A947 (Fig. 1a). No difference in binding affinity to the SMARCA2 and SMARCA4 bromodomains was observed for A947 (Fig. 1b). (SMARCA2 $K_d$ = 93 nM, SMARCA4 $K_d$ = 65 nM) A947 potently degraded SMARCA2 in SW1573 cells with a DC$_{50}$ (the drug concentration that results in 50% protein degradation) of 39 pM and achieving a maximal degradation of 96% at 10 nM (Fig. 1c, d). In contrast, ~28-fold higher concentration of A947 was needed to achieve a DC$_{50}$ on SMARCA4 (1.1 nM), with a maximal degradation of SMARCA4 (92%) being achieved at concentrations approaching 100 nM. This degree of degradative selectivity was maintained independent of the specific isoform of SMARCA2/4 evaluated (Fig. 1e, Supplementary Table 1, Supplementary Fig. 1). Furthermore, A947 exhibited similar selectivity on SMARCA2 degradation over PBRM1 (Fig. 1d). The cellular degradation of SMARCA2 by A947 required both SMARCA2 and VHL binding, as loss of SMARCA2 could be mitigated by the addition of excess free SMARCA2/4 or VHL -binding ligands (Fig. 1f). In addition, a hydroxy-proline diastereomer of A947 expected to attenuate VHL binding (A857), as well as an analog lacking a critical phenol group in the SMARCA2-binding fragment (A858), were largely defective in degrading SMARCA2 in cells; with a negligible impact of A857 on SMARCA2 at the highest concentrations tested (Supplementary Fig. 2). The dependence on ubiquitination and proteosome-mediated degradation was demonstrated by the ability of an inhibitor to the ubiquitin activating enzyme (MLN-7243) and a proteasome inhibitor (MG-132) to block A947-mediated degradation of SMARCA2 (Fig. 1f). A947-mediated cellular degradation of SMARCA2 was rapid, with ~93% loss of the nuclear insoluble pool of SMARCA2 observed within 30 min (Supplementary Fig. 3a–c). Finally, A947 was equally efficient in degrading both the murine and rat orthologs of SMARCA2, as assessed by monitoring the cellular degradation of these orthologs ectopically expressed in cells expressing endogenous human or murine VHL (Fig. 1g, h, Supplementary Fig. 3d).

To more broadly assess the impact of A947 on the ubiquitylome in cells, we carried out quantitative di-glycine reminant profiling by mass spectrometry following treatment of SW1573 cells with a high (500 nM) concentration of A957 to ensure maximal degradation of both SMARCA2/4 (Fig. 2a, Supplementary Data 1). We observed ubiquitination of multiple lysines on both SMARCA2 and SMARCA4, with the strongest ubiquitination on K1450 mapping to the bromodomain of SMARCA2/4. Based on the recently elucidated cryo-EM structure of the BAF complex[27], the majority of the ubiquitination occurred on lysines mapping to the ATPase module and HSA domain, with no ubiquitination observed on very N-terminal lysines that are predicted to be anchored within the core complex. Importantly, we observed no ubiquitination of core BAF complex or accessory proteins. Globally, ubiquitination mediated at this high concentration of A947 in cells was specific to SMARCA2/4. In further support of the selectivity, we quantified degradation at the proteome level by mass spectrometry (Fig. 2b, Supplementary Data 2). SMARCA2/4 and PBRM1 represented the only proteins impacted by A947. Taken together, the data indicate that A947 was highly specific for degrading the expected target proteins at high concentrations.

### A947 can inhibit growth of SMARCA4-mutant NSCLC cells
We next evaluated the impact of A947 on cell proliferation. In SMARCA4$^{mut}$ NCI-H1944 cells, A947 treatment resulted in the dose-dependent inhibition of growth that was dependent upon SMARCA2 degradation, as the VHL and SMARCA2/4 -binding defective analogs (A857 and A858, respectively) were significantly weaker in cells (Fig. 3a). To more broadly assess cellular activity and determine whether the moderate selectivity in degradation translated to selective effects on cell growth, we profiled a panel of lung cancer models characterized by *SMARCA4* mutation status (Fig. 3b, c, Supplementary Data 3). Two additional cell lines that were deficient in SMARCA2/4 expression were included as further controls for any non-specific effect of A947. SMARCA4$^{mut}$ lung cancer cell lines were most sensitive to A947 treatment, with a median IC$_{50}$ of 7 nM across the panel of cell

lines. We did not observe any relationship with the type of *SMARCA4* variant and/or the position of the variant with cellular activity of A947 (Supplementary Fig. 4a). In contrast, SMARCA4[WT] cells were significantly less sensitive to A947, with a median IC$_{50}$ of 86 nM across the panel of models evaluated. A947 treatment had no impact on growth of cells deficient in SMARCA2/4 expression. The difference in cellular growth inhibition between SMARCA4[mut] and SMARCA4[WT] models was not due to differences in the ability of A947 to degrade SMARCA2 ($p = 0.52$, ns) (Supplementary Fig. 4b, Supplementary Data 3). Furthermore, there was a range of IC$_{50}$'s for growth inhibition within SMARCA4[mut] and SMARCA4[WT] models, however we did not observe a direct correlation with growth inhibition and degradation potency of SMARCA2 nor with degradation of SMARCA4 and/or PBRM1 degradation in SMARCA4[WT] models (Supplementary Fig. 4b–d, Supplementary Data 3). This differential in cellular sensitivity to A947 between SMARCA4 mutant and WT cells was also recapitulated upon longer treatment periods in clonogenic growth assays (Supplementary Fig. 4e). A947-mediated degradation resulted primarily in G1 arrest

across SMARCA4[mut] models that was not observed in control cell lines (Fig. 3d). We also did not observe evidence for acute cytotoxicity. At the transcriptional level, A947-mediated SMARCA2 degradation resulted primarily in transcriptional suppression in SMARCA4[mut] cells, consistent with the role of SMARCA2 as a chromatin regulator (Fig. 3e, Supplementary Data 4). Importantly, we observed a strong correlation ($r = 0.53$, $p < 2.2e16$) between the transcriptional changes occurring between A947 treatment compared to inducible SMARCA2 knockdown by shRNA, further supporting the on-target effect of A947 in cells.

## SMARCA2-selective PROTAC, A947, is active in vivo

To address whether A947 is active in vivo, we initially evaluated the pharmacodynamic (PD) effect following a single 40 mg per kg intravenous (IV) dose of A947 in SMARCA4[mut] HCC515 xenografts over a 2 week period (Fig. 4a, Supplementary Fig. 5). In addition to monitoring SMARCA2 protein levels, we evaluated 2 transcriptional target genes that were broadly regulated by SMARCA2 loss across

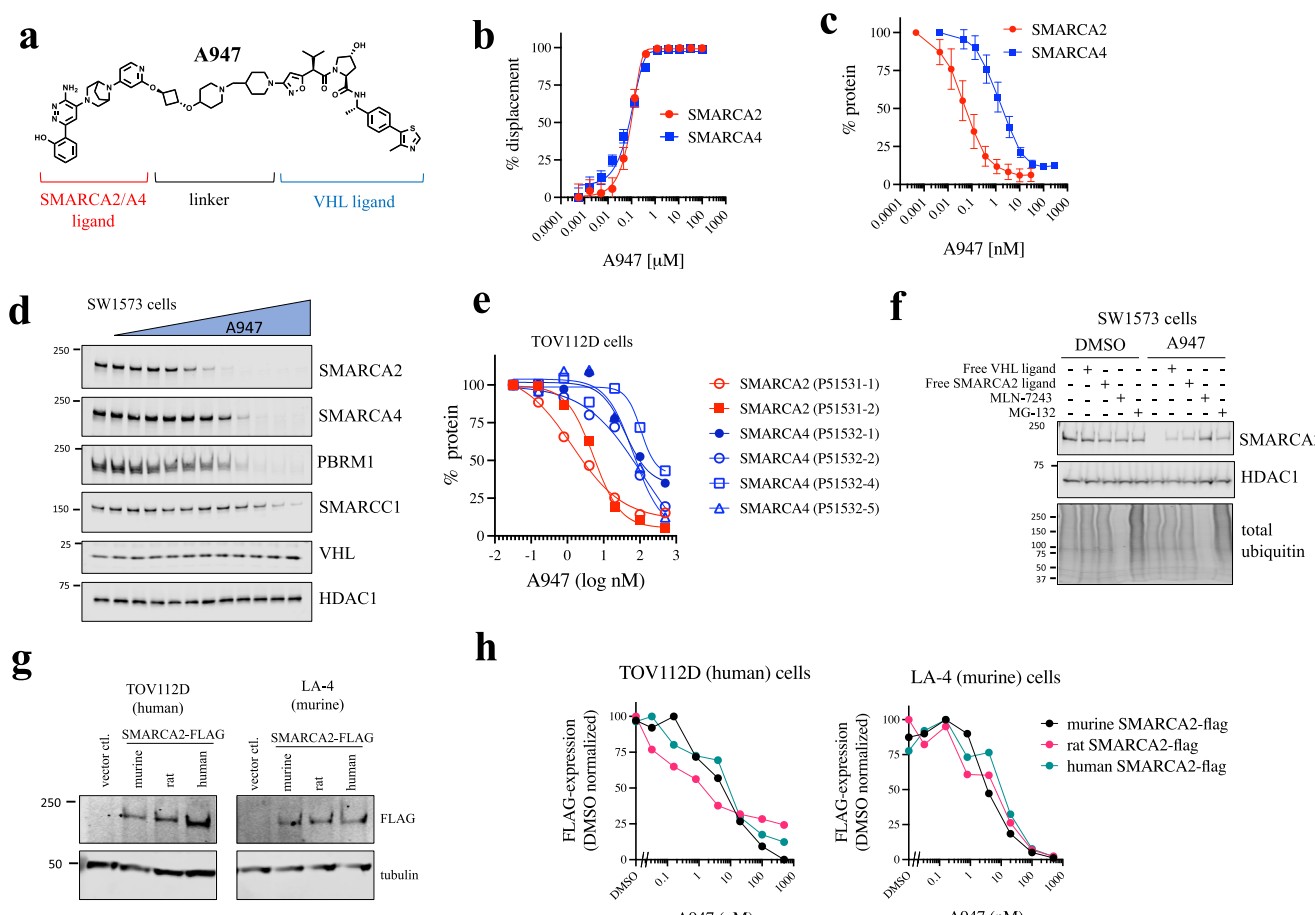

**Fig. 1 | A947 is a potent and moderately selective degrader of SMARCA2.**
**a** Chemical structure of A947. **b** Dose-response curves of A947 displacing a biotinylated SMARCA2/4-binding probe from recombinant SMARCA2 and SMARCA4 bromodomains. Data are presented as mean ± s.d. from 4 replicates.
**c** Quantification of SMARCA2 and SMARCA4 protein levels by In Cell Western™ following 20 h treatment of SW1573 cells. Data are normalized to DMSO control-treated cultures and presented as mean ± s.d. from 7 independent experiments.
**d** Immunoblot analysis of respective proteins following 18 h treatment of SW1573 cells with a dose-response of A947. HDAC1 serves as a loading control. Data is representative of 2 independent experiments. **e** Licor-based quantification of SMARCA2 and SMARCA4 isoforms ectopically expressed in TOV112D cells following 24 h treatment with A947. Data is normalized to DMSO control treated cells.

UniProt identifiers of the respective isoforms are shown in parentheses.
**f** Pretreatment (1 h) of SW1573 cells with 20-fold molar excess of the respective inhibitors can block A947 (500 nM) -mediated degradation of SMARCA2. Total ubiquitin levels serve as a control for MLN-7243 and MG-132. HDAC1 serves as a loading control. **g** Immunoblots demonstrating FLAG-tagged SMARCA2 ortholog expression in human TOV112D and murine LA4 cells. Tubulin serves as a loading control. **h** Licor-based quantification of FLAG-tagged orthologs (human, mouse, rat) of SMARCA2 ectopically expressed in human (TOV112D) and murine (LA4) cell lines upon 24 h treatment with a dose-response of A947. Data are normalized to levels of the respective SMARCA2 ortholog in control (DMSO) lysates. Data in (**f,g**) were confirmed in 3 similar experiments. Source data are provided as a Source Data file.

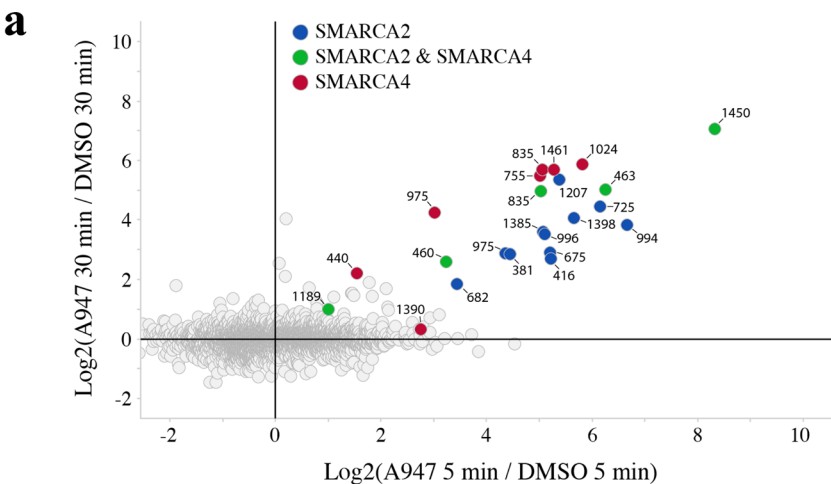

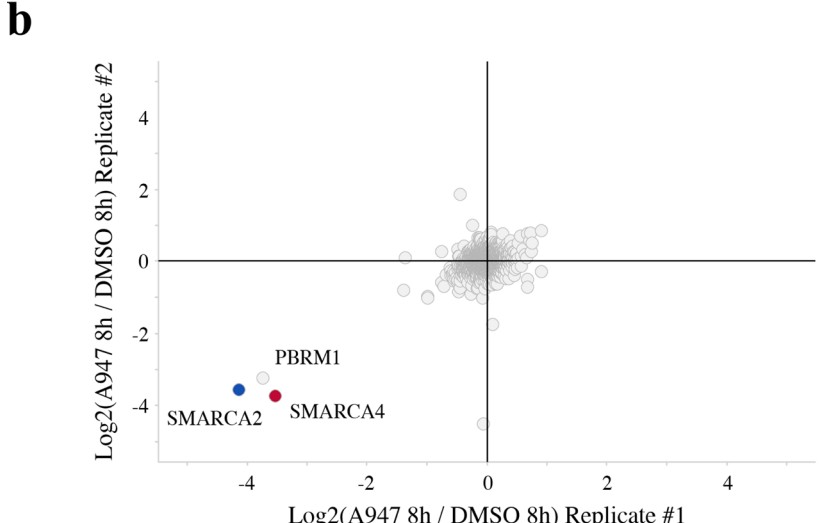

**Fig. 2 | A947-mediated ubiquitination and degradation of SMARCA2/4. a** Global ubiquitylome changes, as assessed by di-glycine reminant mass spectrometry profiling, in SW1573 cells following treatment with 500 nM A947 for 5 and 30 min. $n$ = 5720 unique ubiquitinated peptides were identified and presented as log2 fold change relative to DMSO control cultures. Positions of ubiquitinated lysines in peptides that uniquely map to SMARCA2 (UniProtKB P-51531-1) or SMARCA4 (UniProtKB P-51532-2) are highlighted in blue and red, respectively. Peptides shared between SMARCA2/4 are shown in green, with ubiquitinated lysines mapped to the SMARCA2 sequence. **b** Global proteome assessed by mass spectrometry following 8 h treatment with A947 (100 nM) in SW1573 cells. Data are presented as a log2 fold change in the abundance of the respective proteins in two biological replicates. ~8900 and ~8400 proteins were quantified in replicate 1 and 2, respectively.

SMARCA4$^{mut}$ models, *KRT80* and *PLAU*. Both transcripts are directly regulated by SMARCA2, as previously defined by ChIP-seq and ATAC-seq studies[6]. The IV administration of A947 resulted in rapid reduction (96% reduction by 4 h) in tumor SMARCA2 protein levels and achieved a maximal reduction by 24 h. Loss of SMARCA2 in situ was additionally confirmed by immunohistochemistry (Supplementary Fig. 5b). Decreases in the transcriptional readouts followed, achieving maximal target gene suppression by 96 h post-dose. A slight rebound in SMARCA2 protein levels was observed over the 14 day period as tumor concentrations of A947 decreased; however due to prolonged A947 tumor exposures, these concentrations never reached baseline levels. Differential re-expression of *KRT80* and *PLAU* transcripts was observed following one week after dose administration with *PLAU* mRNA levels paralleling SMARCA2 protein levels.

We subsequently took the dosing regimen of 40 mg per kg every other week i.v. administration of A947 forward into efficacy studies in two different SMARCA4$^{mut}$ lung cancer xenograft models, HCC515 and HCC2302 (Fig. 4b, c). Statistically-significant decreases in tumor growth were observed in both models, highlighted by near complete growth inhibition in the HCC515 model and 60% tumor growth inhibition in the HCC2302 model. Tumor growth inhibition was dependent

upon SMARCA2 degradation and was not due to non-specific effects of the chemical scaffold, as VHL and SMARCA2/4 -binding defective analogs A857 and A858 were not efficacious (Supplementary Fig. 6). Furthermore, tumor growth inhibition was observed in the absence of any appreciable loss in body weight at this dose and regimen, indicating that efficacy was not a consequence of in vivo toxicity (Supplementary Fig. 7a, b). Tumor pharmacodynamic biomarker responses were also measured at end of study; 24 h after administration of a final dose. A947 treatment led to more than a 95% decrease in tumor SMARCA2 protein levels in both models, however a slightly stronger suppression of *KRT80* transcript was observed in the HCC515 model (Fig. 4d). To more extensively address the transcriptional impact of A947, we carried out RNAseq on mRNA isolated from these xenografts and monitored transcripts ($n$ = 412) that were determined from in vitro studies to represent acutely-suppressed and sustained targets of SMARCA2 loss common to both models (Fig. 4e, Supplementary Data 4). Consistent with the differential degree of *KRT80* suppression, we observe slightly stronger suppression of SMARCA2-regulated genes in the HCC515 xenograft model, suggesting that slight differences in the pharmacodynamic effect may underlie the differences in efficacy between these models.

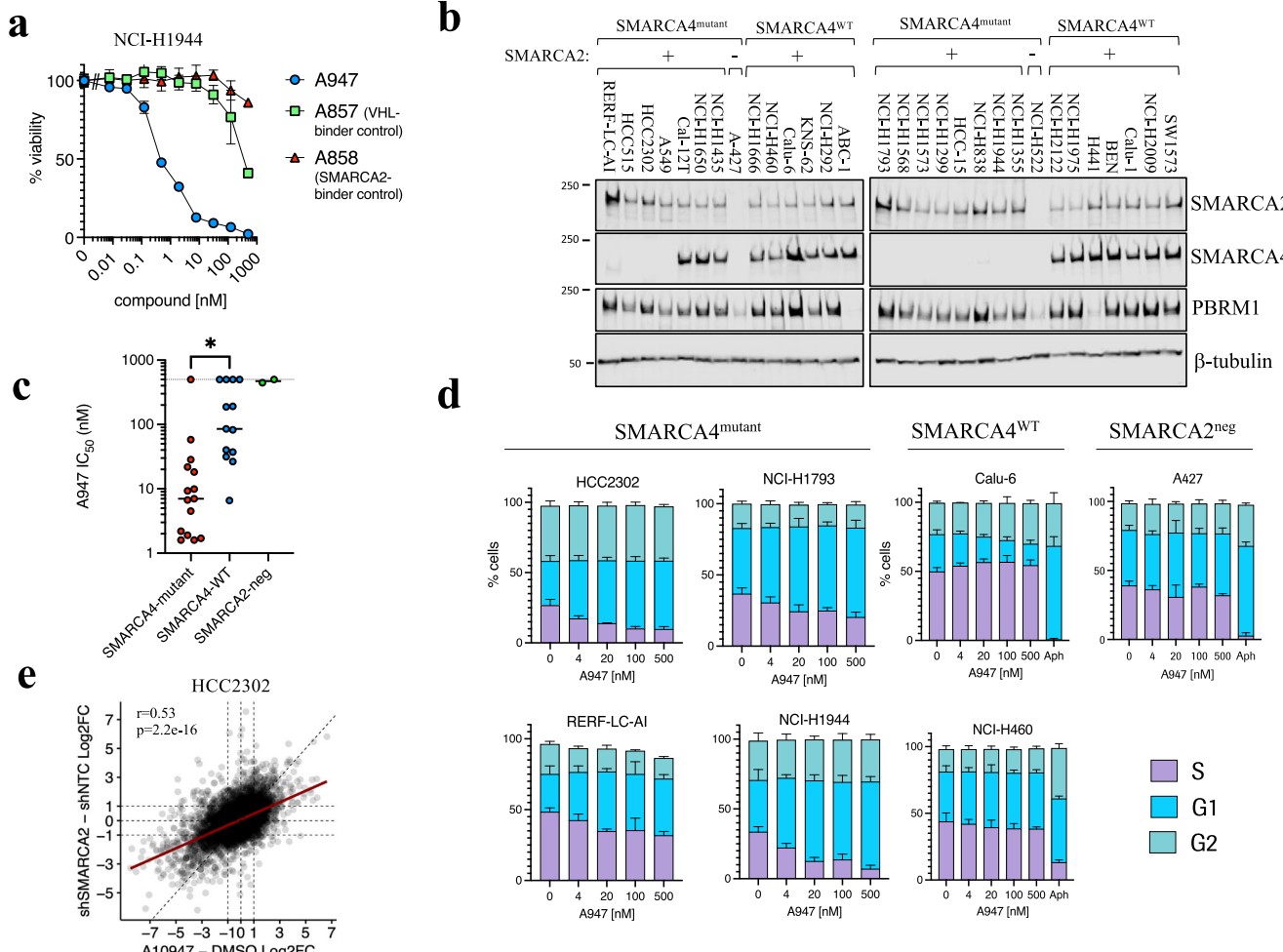

**Fig. 3 | A947-mediated inhibition of cell proliferation in SMARCA4-mutant NSCLC cell lines. a** Viability of NCI-H1944 following 7 days of treatment with a dose response of A947 or binding-defective epimers. Error bars represent mean ± SD from $n = 3$ biologic replicates. **b** Immunoblots of SMARCA2, SMARCA4 and PBRM1 levels across a panel of NSCLC cell lines defined by *SMARCA4* gene mutation status. β-tubulin serves as a loading control. Data are representative of 2 independent experiments. **c** Effect of A947 treatment on the growth of 30 lung cancer cell lines defined by SMARCA4 or SMARCA2 status, represented as the concentration of A947 required to inhibit growth by 50% ($IC_{50}$) following 7 days of treatment. Individual cell line $IC_{50}$'s were determined from $n = 3$ biologic replicates. Median $IC_{50}$'s across models defined by mutational status are indicated by the black line. Significance was assessed by a two-tailed, Mann Whitney test. Asterick indicates

$p = 0.0003$ **d** Cell cycle distribution following 48 h treatment of a dose response of A947 across 4 SMARCA4-mutant, 2 SMARCA4-wt and one SMARCA2-deficient NSCLC cell lines. Aphidicolin (Aph, 1 μM) treatment served as a control to block entry into S phase for the SMARCA4-wt and SMARCA2-deficient models. Error bars represent mean ± SD from $n = 3$ biologic replicates. **e** Log$_2$-transformed fold change in mRNA expression values, as measured by RNA-seq, in HCC2302 cells treated for 96 h with A947 compared to control DMSO treated cultures (x-axis), as well as HCC2302-shSMARCA2 cells treated with doxycycline for 168 h compared to HCC2302 cells expressing a non-targeted control shRNA (shNTC). RNAseq data is representative of triplicated cultures. The correlation was calculated by the Pearson coefficient. $p = 2.2\mathrm{e}{-16}$. Source data are provided as a Source Data file.

In order to address whether the tumor growth inhibition observed in SMARCA4$^{mut}$ models was due to a tumor cell autonomous effect of SMARCA2 degradation, we evaluated A947 administration in the SMARCA4$^{wt}$ Calu-6 xenograft model (Fig. 4f). Applying the same dose and regimen as used in the SMARCA4$^{mut}$ xenograft studies, A947 did not result in tumor growth inhibition in SMARCA4$^{wt}$ Calu-6 xenografts despite achieving greater than 95% degradation of SMARCA2 protein (Fig. 4g, Supplementary Fig. 8a). Moderate degradation of SMARCA4 and PBRM1 was observed at the 24 h post-last dose timepoint, with 58% and 57% decreases respectively. Although degradation selectivity cannot be addressed in SMARCA4$^{mut}$ models due to the deficiency in human SMARCA4, we were able to monitor murine SMARCA4 protein levels within the tumor microenvironment of HCC515 and HCC2302 xenografts (see Supplementary Fig. 5b for in situ confirmation of stromal SMARCA4 signal). Analogous to the Calu-6 xenografts, 57% and 69% reductions in murine SMARCA4 levels were observed in these studies, suggesting that murine SMARCA4 is degraded similarly to

human SMARCA4 (Supplementary Fig. 8b, c). Hence, the moderate selectivity for SMARCA2 over SMARCA4 degradation observed in vitro for A947 translated to a moderate degradative selectivity in vivo at this dose and regimen. Taken together, the data are supportive of a tumor cell intrinsic effect of SMARCA2 degradation and provide pharmacologic support of the synthetic lethal interaction.

## Synergistic combination of A947 with MCL1 inhibitors

Although SMARCA2 degraders would have the potential to be developed as single agents in the clinic to treat SMARCA4$^{mut}$ cancers, we have begun to address whether rational and ubiquitously active pharmacologic combinations exist for SMARCA2 degraders in SMARCA4$^{mut}$ cancers. To assess combination effects, we screened a library of 723 experimental and clinically approved agents in combination with A947 across 4 SMARCA4$^{mut}$ lung cancer cell lines (Fig. 5a, Supplementary Data 5). Although A947 treatment sensitized to unique compounds in a given model, MCL1 inhibition represented the only

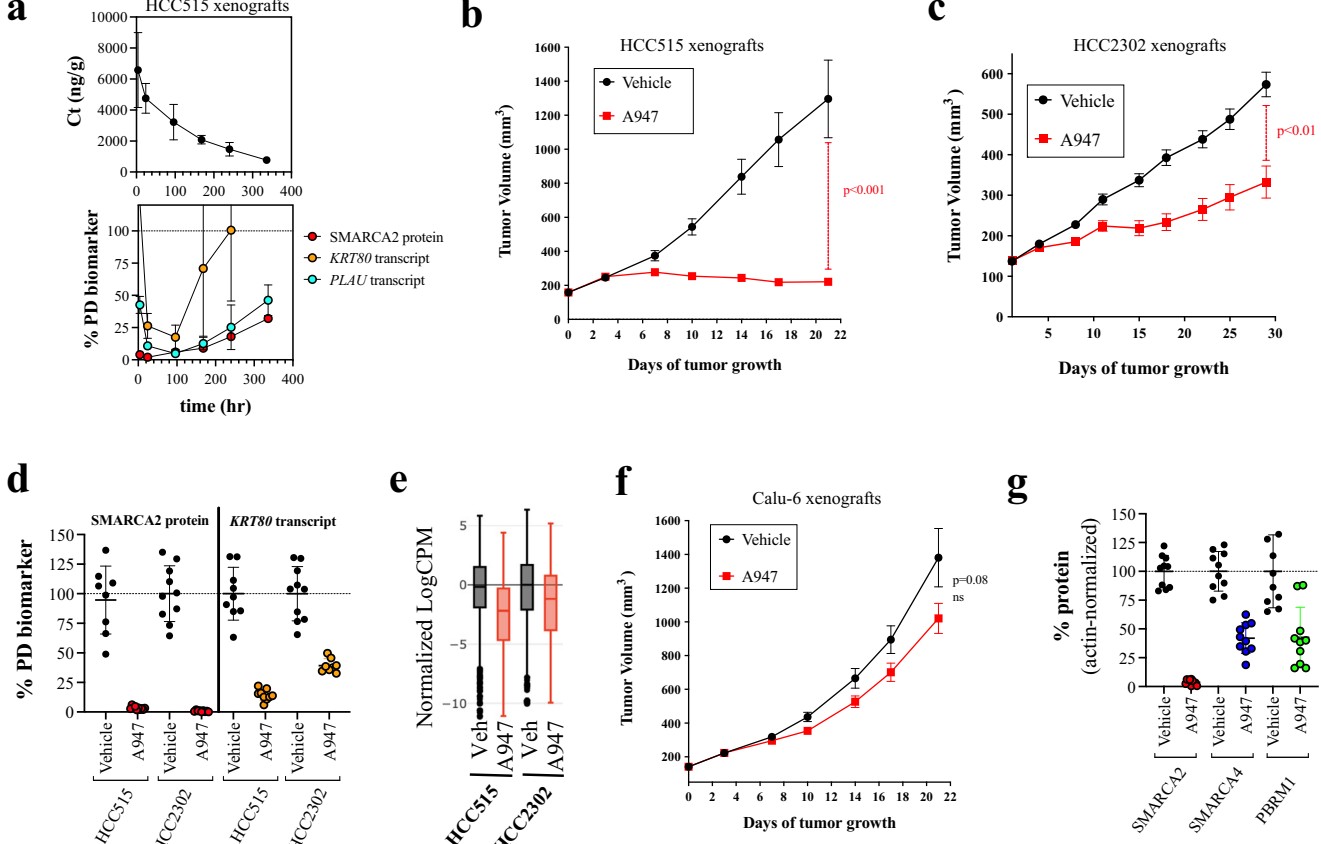

**Fig. 4 | A947 is selectively efficacious in SMARCA4-mutant NSCLC xenograft models. a** Tumor concentration (upper graphic) and pharmacodynamic biomarker responses (lower graphic) monitored over a 2 week period in HCC515 xenograft tumors following a single-dose, intravenous (i.v.) administration of A947 (40 mg/kg). Tumor levels of SMARCA2 protein were quantified by ImageLab from Western blots and normalized to a loading control protein (β-actin). Tumor levels of the mRNA transcripts, *KRT80* and *PLAU*, were quantified by Taqman. Data are presented relative to levels in untreated tumors and represented as mean ± s.d from *n* = 5 animals per timepoint. **b, c** Tumor volume in mice harboring either SMARCA4-mutant HCC515 (**b**) or HCC2302 (**c**) xenografts following administration of A947 (40 mg/kg, i.v.) or vehicle control. Data are presented as mean ± s.e.m. (*n* = 10 mice/group). Statistical significance was assessed by a two-sided, unpaired Student *t* test. *p* = 0.0002; *p* = 0.0049 (**c**). **d** Pharmacodynamic biomarker levels tumors collected at end of study from animals treated in (**b** and **c**) (SMARCA4 protein: HCC515-vehicle (*n* = 8), HCC515-A947 (*n* = 10), HCC2302-vehicle (*n* = 10), HCC2302-A947 (*n* = 9); KRT80 transcript: HCC515-vehicle (*n* = 9), HCC515-A947 (*n* = 9), HCC2302-vehicle (*n* = 10), HCC2302-A947 (*n* = 8). Mice received a final dose of A947 24 h prior to tumor collection. SMARCA2 protein and *KRT80* mRNA levels were quantified as

in (**a**) and presented relative to levels in the respective vehicle-treated tumors. **e** Log-normalized counts per million reads (CPM) of SMARCA2 target genes determined by RNAseq and presented relative to levels in vehicle-treated tumors. A consensus set of SMARCA2-regulated genes (*n* = 412) were defined based upon exhibiting acute (24 h) and sustained (96 h) suppression following A947 treatment of both models in vitro, as further described in the methods section. Boxes span from quartile 1 (Q1; 25th perentile) to quartile 3 (Q3; 75th pertentile), with medians (50th percentile) represented by center lines. Whiskers span ± 1.5 times the interquartile range (Q3–Q1) and outliers are represented as dots. **f** Tumor volume in mice harboring SMARCA4 wild-type Calu-6 xenografts following administration of A947 (40 mg/kg, i.v.) or vehicle control. Data are presented as mean ± s.e.m. (*n* = 10 mice/group). Potential significance was assessed by a two-sided, unpaired Student *t* test. n.s. nonsignificant. **g** Levels of SMARCA2, SMARCA4 and PBRM1 protein in Calu-6 tumors (*n* = 10 for each group) were quantified by ImageLab from Western blots and normalized to a loading protein control (β-actin). Calu-6 tumors were collected at the end of study from mice treated in (**f**) and mice received a final dose of A947 24 h prior to tumor collection. Source data are provided as a Source Data file.

combination exhibiting strong sensitization with A947 in more than one SMARCA4^mut model (3 of the 4 models), as well as with multiple inhibitors. To confirm and extend this observation, we carried out a matrix titration of 2 separate MCL1 inhibitors (AMG-176 and S63845) with A947 and evaluated synergistic growth inhibition based upon the Bliss independence model in 5 SMARCA4^mut models (Fig. 5b, Supplementary Fig. 9). In all cases, A947-mediated SMARCA2 degradation exhibited a synergistic interaction with MCL1 inhibition. Synergistic growth inhibition was not observed in 2 SMARCA4^WT models (Supplementary Fig. 9). Given the cytostatic effect of A947 treatment as a single agent and the anti-apoptotic function of MCL1, we evaluated whether the synergy was due to the ability of the combination to induce apoptosis. Indeed, live cell imaging of activated caspase 3/7 in SMARCA4^mut cells revealed that the combination was able drive these cells toward apoptotic cell death (Fig. 5c). These data, combined with

previously published work leveraging a genetically-engineered degron tagging approach[28], support a potential combination of SMARCA2 degraders with MCL1 inhibitors.

## Discussion
In this study, we report a moderately selective SMARCA2-targeting PROTAC, A947, that is active both in vitro and in vivo in SMARCA4^mut NSCLC models. A947 exhibited a 28-fold selectivity in degradation of SMARCA2 over SMARCA4 and notably exhibited no unexpected off-target effects at high concentration in both global ubiquitinome and proteome studies in cells. The ability of A947 to phenocopy the effect of inducible shRNA-mediated knockdown of SMARCA2 on transcriptome expression further supported the molecule's on target effect. A947 differs from a previously reported SMARCA2/4-targeting PROTAC, ACBI1[26]. Despite differences in the models tested, A947

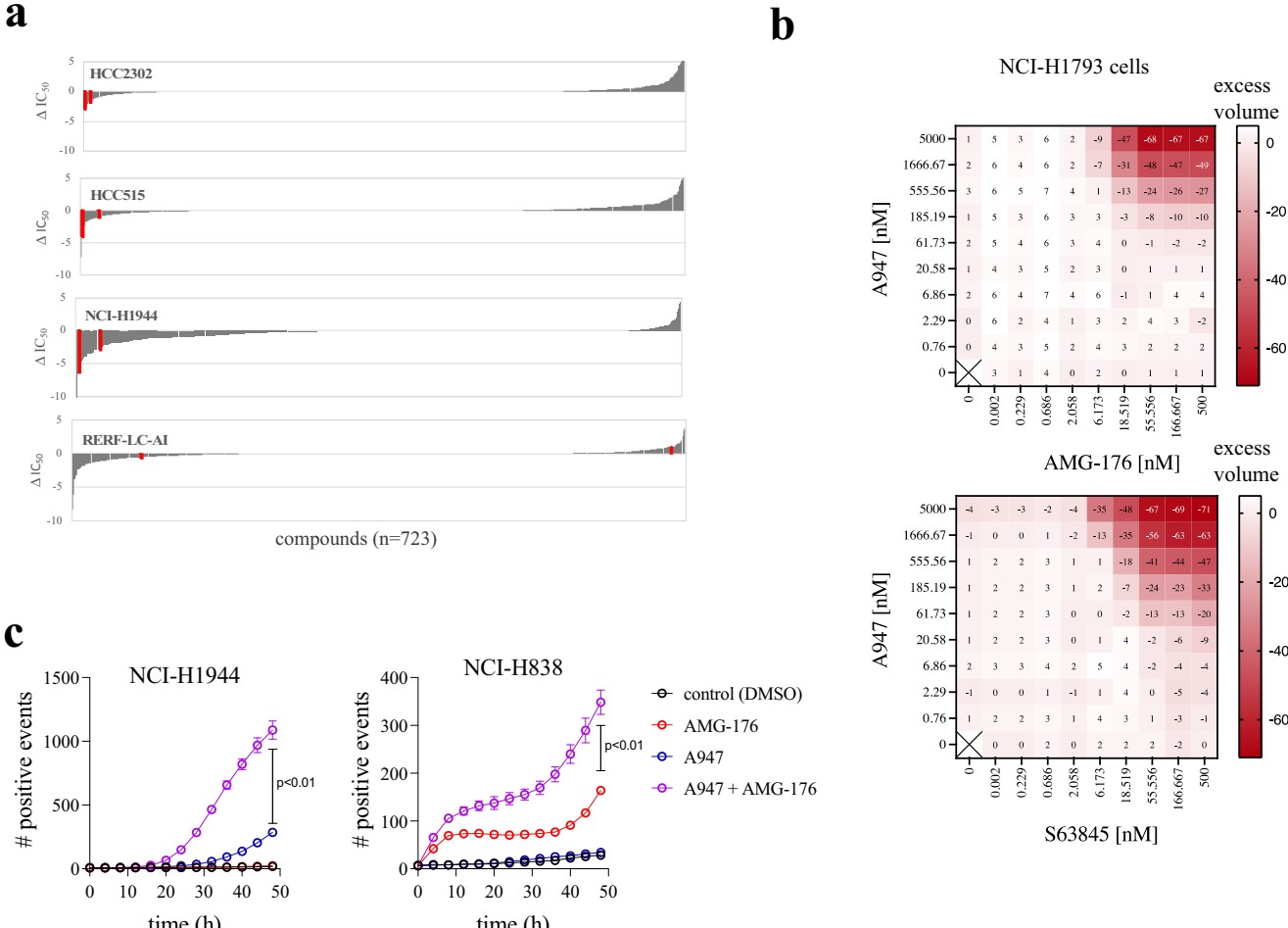

**Fig. 5 | A947-mediated SMARCA2 degradation synergizes with MCL1 inhibition. a** Cell viability in 4 SMARCA4 mutant NSCLC cell lines measured across a small-molecule library of 723 compounds screened as a dose-response in the presence or absence of 100 nM A947 for 5 days. Data are plotted as the difference in the concentration required to inhibit growth by 50% ($\Delta IC_{50}$) upon −/+ A947 treatment. Two separate MCL1 inhibitors are annotated in red. **b** Treatment of a representative SMARCA4-mutant cell line (NCI-H1793) with a 9 × 9 matrix titration of A947 with the MCL1 inhibitors, AMG-176 (upper plot) and S63845 (lower plot). Heatmaps depict activity in excess of the Bliss independence model to describe synergistic drug interactions (excess volume). **c** Live cell monitoring of apoptosis in SMARCA4-mutant NCI-H1944 (left graph) and NCI-H838 (right graph) cells grown in the presence of Caspase-3/7 Green Dye upon treatment with 100 nM A947 and/or 1 µM AMG-176. Data are presented apoptotic object count (mean ± s.d) in triplicate cultures. Significance was calculated by a two-, unpaired *t*-test; *p* = 0.0078 (NCI-H1944), *p* = 0.0002 (NCI-H838). Source data are provided as a Source Data file.

exhibited greater SMARCA2 degradation potency (39pM $DC_{50}$ v. 6 nM $DC_{50}$) and selectivity over SMARCA4 (28.2-fold v. 1.8-fold shift in $DC_{50}$)[26]. This 28-fold selectivity in cellular degradation achieved with A947 translated to selective tumor growth inhibition both in vitro and in vivo in SMARCA4[mut] cancers compared to SMARCA4[wt] cell line models. Pharmacodynamic profiling revealed that moderate in vivo degradation (~60%) of SMARCA4 could occur along with the associated strong SMARCA2 degradation (>95%), however the former activity was not sufficient to either drive efficacy in a SMARCA4[wt] model nor to impact tolerability at the dosing regimen utilized. Taken together, these data provide pharmacologic support of this paralog-mediated, synthetic lethality; and provide an early indication that a potent SMARCA2 degrader with ~30-fold selectivity could be safely delivered.

An important aspect of this effort has been the ability to exploit non-selective SMARCA2/4 binding ligands to achieve selective cellular degradation in the context of the PROTAC. However, despite our ability to achieve selective cellular degradation, the mechanism underlying selectivity remains to be elucidated. Differences in ternary complex affinity have been attributed to several reported examples of selective degraders that utilize ligands with promiscuous binding

properties[23,24] and ternary complex formation has been previously observed with non-selective SMARCA2/4 PROTACs[26]. However, it remains to be determined whether fine differences in cooperativity between SMARCA2 and the VHL complex (VCB: VHL-ElonginC-ElonginB) compared to SMARCA4/VCB underlie the difference in cellular degradation between the SMARCA2/4 paralogs. In vitro measurements of ternary complex formation have relied on the use of isolated bromodomains, which may not simulate the physiologically relevant substrate:PROTAC:VCB interface in the context of the native BAF complex. Selectivity could also be potentially driven by cell-intrinsic factors, as opposed to the biophysical properties of the ternary complex. Nevertheless, understanding the biochemical and/or cellular mechanism of degradation selectivity of A947 remains an active area of research.

Due to their high molecular weight and poor physiochemical properties, the oral delivery of VHL-based PROTACs can be problematic. Preclinical efficacy studies have generally required intraperitoneal or subcutaneous administration of VHL-PROTACs at high doses and high frequencies, with few exceptions[29]. While the poor physiochemical properties of VHL-based PROTACs complicate their oral delivery, opportunities exist for alternative parenteral routes of

administration[30]. Intravenous (IV) dosing offers some obvious advantages, such as the lack of physiological barriers to absorption, but requires solubility consistent with the target dose and an intravenous half-life ($T_{1/2}$) sufficiently long to satisfy the PKPD requirements and a likely intermittent dosing schedule. Prolonged IV $T_{1/2}$ can be achieved by increasing PROTAC affinity to the lipid and phospholipid cell components (either via addition of a positive charge or increased lipophilicity), while decreasing or maintaining intrinsic metabolic stability[31,32]. The amine containing linker contained in A947 promotes high solubility in the IV formulation, as well as moderate CL (<16 ml/min/kg) and high affinity for body tissues in rodents, resulting in $VD_{ss} > 6$ L/kg and $T_{1/2} > 6$ h. Interestingly, in rodents the residence time of A947 in tumors (and presumably in other body tissues) appears to be much higher than in plasma, suggesting that dissociation from tissues is a slower process compared to metabolic elimination from plasma (Fig. 4a, Supplementary Fig. 5a). While these observations were not mechanistically investigated, they are consistent with intracellular lysosomal trapping, a phenomenon frequently observed for lipophilic basic amines[33].

Overall, our data may have future clinical implications, offering a potential therapeutic option for patients harboring SMARCA4[mut] cancers. Although A947-mediated degradation of SMARCA2 results primarily in cytostasis that is consistent with prior genetic perturbation studies[12], the overall depth of single-agent efficacy in the context of an immune-competent animal and/or upon prolonged dosing remains to be determined. Nevertheless, we considered whether rational drug combinations exist as a means to potentiate the tumor cell intrinsic, cellular activity of A947. We specifically sought to identify drug synergies that were broadly active and not necessarily cell context specific. Through pharmacologic profiling, we determined that inhibition of the BCL2-family pro-survival protein, MCL1, could synergize with SMARCA2 degradation across multiple SMARCA4[mut] models. Interestingly, MCL1 was additionally identified in a genome-wide CRISPR knockout screen to identify sensitizers to SMARCA2 loss in a genetically engineered system whereby SMARCA2 was endogenously tagged with the SMASh degron to enable degradation with the NS3 protease inhibitor, asunaprevir[28]. Drug sensitization was observed with additional anti-apoptotic drugs including BCL-XL and IAP antagonists, however these effects were cell line specific. Combined with the prior report, these data may indicate the broader utility combining MCL1 antagonists with a SMARCA2 PROTAC. Multiple MCL1 inhibitors are under early clinical investigation and are being considered in combination with other therapeutics in solid tumors as a means to lower the apoptosis threshold[34].

## Methods

This study complies with all relevant ethical regulations. Animals were maintained in accordance with the Guide for the Care and Use of Laboratory Animals. Genentech is an AAALAC-accredited facility and all animal activities in the research studies were conducted under protocols approved by the Genentech Institutional Animal Care and Use Committee (IACUC).

### Chemical compounds

A947, A857, A858 and the free VHL ligand (A2702) were prepared by Arvinas, Inc., with detailed chemical synthesis described at the end of this Methods section. The chemical compounds MLN-7243 and MG-132 were obtained from SelleckChem. The SMARCA4/2 bromodomain inhibitor (example 47, WO2016138114) was used in competition studies (Fig. 1f).

### Antibodies

The following antibodies were utilized: SMARCA2 (Cell Signaling, 11966, dilution 1:2000), SMARCA4 (Abcam, ab110641, dilution 1:1000), PBRM1 (Bethyl Labs, A301-591A, dilution 1:1000), SMARCC1 (Cell Signaling, 11956, dilution 1:1000), HDAC1 (Cell Signaling, 34589, dilution 1:1000), VHL (Cell Signaling, 68547, dilution 1:1000), FLAG (Sigma, F3165, dilution 1:1000), Lamin A/C (Cell Signaling, 4777, dilution 1:1000), α–Tubulin (Sigma, T6074, dilution 1:1000), β-Tubulin (Cell Signaling, 2128, dilution 1:1000), β-Actin (Cell Signaling, 3700, 1:1000 and 4970, dilution 1:1000) and total ubiquitin (Cell Signaling, 3936, dilution 1:1000).

### Cell lines and cell culture

Cell lines were obtained from the following sources (indicated in Supplementary Data 3): American Type Culture Collection (ATCC), Japanese Collection of Research Bioresources Cell Bank (JHSF), Deutsche Sammlung von Mikroorganismen und Zellkulturen (DSMZ), Riken, or licensed from UT Southwestern (UTSW). 293 T cells were obtained from ATCC. Cells were maintained in RPMI1640 supplemented with 10% Fetal Bovine Serum (FBS) and 2mM L-Glutamine, under 5% CO2 at 37 °C, with the exception of SW1573 (DMEM) and Calu-6 (EMEM). Cell line identity was verified by high-throughput single nucleotide polymorphism (SNP) genotyping using Illumina Golden Gate multiplexed assays. SNP profiles were compared to SNP calls from internal and external databases to determine or confirm ancestry. All cell lines tested negative for mycoplasma contamination prior to storage/use at our institute.

### AlphaLISA®

AlphaLISA® studies were carried out using histidine (His)-tagged recombinant human SMARCA2 (aa.1377-1486; NP_620614) or SMARCA4 (aa. 1448-1575; NP_003063) proteins expressed in *Escherichia coli*. Compounds were diluted with 3-fold dilutions (11-point) in 96-well plates with a top concentration of 10 mM in 100% DMSO. Compounds were further diluted tenfold in Alpha LISA buffer consisting of 50 mM HEPES (Life Technologies 15630-080), 50 mM NaCl (Sigma BCBW5699), 69uM Brij (Sigma SLBM8986V), and 0.1 mg/ml BSA (Sigma A7906-100G) brought up to 100 mL final volume in water (Sigma RNBG4333), resulting in a top concentration of 1 mM in 10% DMSO. 3uL of this dilution was spotted into 384 well plate(s) (Perkin Elmer 6007290), final reaction volume of 30 uL, with compounds having a final top concentration of 0.1 mM. The final 30 ul reaction consisted of the following components: Compound, 7 nM His-SMARCA2 (or His-SMARCA4), 20 nM SMARCA2/SMARCA4 Biotin Probe (example 248, WO2016138114), 1:400 Dilution of anti-His Alpha-LISA Acceptor Beads (Perkin Elmer AL128M), and 1:400 Dilution of Streptavidin Alpha-LISA Donor Beads (Perkin Elmer 6760002). To make a working stock of His-SMARCA2/SMARCA4 and biotinylated probe, 1.7 uL of 60uM His-SMARCA2 (or His-SMARCA4) stock and 30 uL of 10 uM biotinylated probe (diluted in Alpha LISA buffer) was added to 20 mL Alpha LISA buffer to give final concentrations of 17.5 nM protein and 50 nM probe. Mixtures were then incubated at room temperature for 5–10 min. Then 12 uL of protein/probe mixture was added to each reaction well in 384 well plate containing compounds (or probe only for background control wells). Plates were incubated for 10–15 min at room temperature. Anti-His6x Acceptor beads were diluted 100× in Alpha-LISA Buffer protected from light and 7.5 uL added to each well. Plates were incubated 10–15 min at room temperature protected from light. Streptavidin donor beads were diluted 100X in Alpha-LISA Buffer protected from light and 7.5 uL added to each well. Plates were incubated for at least 15 min (no more than 4 h) at room temperature protected from light and plates read at 615 nm on a micro-plate reader. For data analysis, the averages of control wells (probe + protein only max signal and probe only background control) were calculated and emission values at 615 nm were used to calculate percent displacement values using this formula:

$$(probe/protein − probe/protein/compound)/(probe/protein − probe\ only) * 100 = \% \ displacement$$

Other values, such as mean and standard deviation, were calculated using GraphPad Prism software package.

### In-cell Western

SW1573 cells were seeded at 8000/well in 96-well black/clear-bottom plates with 180ul DMEM containing 1% pen-strep, 1% HEPES and 10% FBS and incubated overnight at 37 °C to allow adherence. The next morning cells were treated with 20 uL of 10× compound and incubated for an additional 20 h. Cells were then washed with ice cold DPBS, then 50 uL of ice cold 4% PFA (Electron Microscopy Sciences 15711)/DPBS was added, and plates were then incubated at RT for 20 min. PFA was then removed and 200 uL of TBS-T containing 0.5% Triton X 100 (Sigma T8787) was added. Plates were incubated at RT for 30 min. TBS-T/Triton X was then removed, 50 uL of Li-cor blocking solution (Li-Cor 927-50003) was added, and plates were incubated at RT for 1 h. Blocking solution was removed, 50 uL of Li-Cor blocking solution containing primary antibody to SMARCA2 (1:2000) and α-Tubulin (1:2000) or SMARCA4 (1:1000) and α-Tubulin (1:2000) was added, and plates were incubated at 4 °C overnight. The next day, plates were washed 3× with TBS-T and 50uL of secondary antibody cocktail in Li-Cor blocking solution was added (IRDye 800CW Goat anti-rabbit IgG, Li-Cor 926-32211 and IRDye 680RD Goat anti-mouse IgG, Li-Cor 926-68070; both secondary antibodies are diluted 1:5000). Plates were incubated at RT for 1 h protected from light. Then plates were washed twice with TBS-T and excess liquid removed. Plates were read on the Li-Cor Odyssey with default intensity setting of 5.0 for both channels. Li-Cor images were analyzed with the in-cell Western feature of Image Studio Lite. Assays were run with technical duplicates and multiple biological replicates (>2) with error calculated with GraphPad prism using 95% confidence interval.

### Immunoblotting (in vitro samples)

Following treatment of cells as indicated, cells were lysed in RIPA buffer containing 0.5 M NaCl, then homogenized for 3 min at speed 10 (NextAdavance, Bullet Blender[R] 24). Proteins (12 ug or 18 ug) were resolved on 4–12% Bis-Tris or 3-8% Tris-Acetate gels and transferred to nitrocellulose membranes by iBlot. Membranes were incubated overnight with primary antibodies as indicated. IRDye[R] -conjugated secondary antibodies were used to bind primary antibodies and images were visualized on the Odyssey Imager (LI-COR).

### SMARCA2/4 isoform and ortholog expression

The pLenti6.3 vector system was used for all ectopic expression experiments. All DNA constructs were generated with a C-terminal FLAG tag. The human SMARCA2 and SMARCA4 isoform sequences are annotated in Supplementary Table 1. For SMARCA2 orthologs, the following sequences were utilized: human (NM_139045.2), rat (XM_006231227.3), and mouse (NM_011416.2). The packaging and envelope vectors, Δ8.91 and VSV.G were cotransfected with pLenti6.3-based constructs into 293 T cells by using lipofectamine 2000 (Invitrogen). Media containing lentiviral particles was collected 3 days post transfection, filtered through 0.45 μm filters. Cells (TOV112D (human) and/or LA-4 (murine)) were transduced with pLenti6.3- constructs particles, and selected with 8 μg/mL blasticidin (Gibco, A1113903) 3 days after transduction.

### Generation of inducible SMARCA2 shRNA cells

Inducible SMARCA2 knockdown cell lines were generated using shRNAs directed against SMARCA2 tandemly delivered in a modified pBH1.2 piggy-bac system (Smarca2_iKD_dual62_pBH1.2). The following shRNAs were utlilized: shRNA6:GATCCGTCTCGTCGAGCAATCAT TTGGTTGTAGTGAAATAtATATTAAACAACCAAATGATTGCTCGACGT TACGGTAC and shRNA2:GATCCGTCTGACTGTTCACGTTCATCCTGGT AGTGAAATAtATATTAAACCAGGATGAACGTGAACAGTCTTACGGTAC.

pBO (Piggybac transposase) were co- transfected with Smarca2_iKD_dual62_pBH1.2 into HCC2302 cells by using lipofectamine 2000 (ThermoFisher Scientific). Cells were selected with 2 μg/mL puromycin (Gibco) 3 days after transfection and subsequently subcloned.

### Sample preparation for quantitative proteome and ubiquitylome analysis

Cells were lysed on plate using a lysis buffer consisting of 9 M urea, 50 mM HEPES (pH 8.5), and complete-mini (EDTA free) protease inhibitor (Roche). Protein concentrations were then estimated by BCA assay (ThermoFisher Pierce, Rockford, IL). Disulfide bonds were reduced with 5 mM DTT (45 min, 37 °C), followed by alkylation of cysteine residues by 15 mM IAA (30 min, RT Dark), and finally capped by the addition of 5 mM DTT (15 min, RT Dark). Proteins were then precipitated by chloroform/methanol precipitation and resuspended in digestion buffer (8 M urea, 50 mM HEPES pH 8.5). Samples were diluted to 4 M urea before initial protein digestion was performed by the addition of 1:100 LysC followed by incubation at 37 °C for 3 h. Samples were then diluted to 1.5 M urea with 50 mM HEPES (pH 8.5) before the addition of 1:50 Trypsin and incubation overnight at room temperature. Peptide mixtures were acidified and desalted via solid phase extraction (SPE; SepPak - Waters, Boston, MA).

For global proteome analysis, peptides were resuspended in 200 mM HEPES (pH 8.5) and a 100 μg aliquot of peptides was mixed with tandem mass tags (TMT or TMTpro, ThermoFisher Pierce, Rockford, IL) at a label to protein ratio of 2:1. After 1 h of labeling, the reaction was quenched by the addition of 5% hydroxylamine and incubated at room temperature for 15 min. Labeled peptides were then mixed, acidified, and purified by SPE. Samples were separated by off-line high pH reversed-phase fractionation using an ammonium formate based buffer system delivered by an 1100 HPLC system (Agilent). Peptides were separated over a 2.1 × 150 mm, 3.5 μm 300Extend-C18 Zorbax column (Agilent) and separated over a 75-minute gradient from 5% ACN to 85% ACN into 96 fractions. The fractions were concatenated into 24 samples of which 12 or 24 were analyzed for proteome quantification. Fractions were concatenated by mixing different parts of the gradient to produce samples that would be orthogonal to downstream low pH reversed phase LC-MS/MS. Combined fractions were dried, desalted by SPE, and dried again.

For ubiquitylome analysis, peptides were resuspended in 1X detergent containing IAP buffer (Cell Signaling Technology), cleared by high speed centrifugation (18,000 × g, 10 min) and enriched using an automated procedure previously described[35]. Enriched ubiquitinated peptides were prepared for multiplexed quantitative analysis as previously described except that TMTpro reagents were used[36]. All six fractions were analyzed by LC-MS/MS.

### Quantitative mass spectrometry and data analysis

Quantitative mass spectrometry analysis was performed on an Orbitrap Fusion Lumos or Orbitrap Eclipse mass spectrometer (ThermoFisher, San Jose, CA) coupled to a Waters NanoAcquity (Waters, Milford, MA) or Thermo Ultimate 3000 RSLCnano ProFlow (ThermoFisher, San Jose, CA) HPLC. Peptides were separated over a 100 μm × 250 mm PicoFrit column (New Objective) packed with 1.7 μm BEH-130 C18 (Waters, Milford, MA) at a flow rate of 450 or 500 nL/min or over a 25 cm IonOpticks Aurora column (IonOpticks, Fitzroy, Australia) at 300 nL/min for a total run time of 180 min. The gradient started at 2−5% B (98% ACN, 1% FA) and ended at 30% B over 140 min and then to 50% B at 160 min.

Peptides were surveyed via Orbitrap FTMS1 analysis (120,000 resolution, AGC = 1 × 106, maximum injection time [max IT] = 50 ms) and the most intense 10 peaks were selected for MS/MS ensuring that any given peak was only selected every 35 or 45 s (tolerance = 10 ppm). For all runs, "one precursor per charge state" was ON. For data

collected on the Eclipse mass spectrometer, Advanced Precursor Detection (APD), FAIMS (CVs = −50, −70), and the Precursor Fit Filter (70% fit and 0.5 fit window) were employed.

For peptide identification, precursors were isolated using the quadrupole (0.5 Th window), fragmented using CAD (NCE = 35 for TMT and NCE = 30 for TMTpro) and analyzed in the ion trap using a Turbo speed scan (AGC = $2 \times 10^4$, maxIT = 100 ms) for proteome analysis or an Orbitrap scan at 15,000 resolution (AGC = $7.5 \times 10^4$, maxIT = 200 ms) for ubiquitylome analysis. A real-time database search was utilized for both proteome and ubiquitylation quantification on the Eclipse mass spectrometer. The real-time database search performed an in silico trypsin digest with full specificity and 1 missed cleavage and used concatenated decoy proteins to calculate FDR in real time. The precursor PPM tolerance was set to 10 ppm and the real-time search static and dynamic modifications matched the search parameters below. For quantitative SPS-MS3 analysis, the top 8 ions were simultaneously selected (synchronous precursor selection – SPS, AGC = $1.5 \times 10^5$ or $3.0 \times 10^5$ [proteome] or $4.0 \times 10^5$ [ubiquitylome], max IT = 150 ms [proteome] or 400 ms [ubiquitylome]) and fragmented by HCD (NCE = 55 [TMT] or 40 [TMTpro]) before analysis in the Orbitrap (resolution = 50,000). The mass spectrometry proteomics data have been deposited to the ProteomeXchange Consortium via the PRIDE partner repository with the dataset identifier PXD036865.

All mass spectrometry data was searched using Mascot against a concatenated target-decoy human database (downloaded June 2016) containing common contaminant sequences. For the database search a precursor mass tolerance of 25 ppm (TMTpro) or 50 ppm (TMT), fragment ion tolerance of 0.5 Da (TMTpro) or 0.8 Da (TMT), and 1–2 (proteome) or 3 (ubiquitylome) missed cleavages. Carbamidomethyl cysteine (+57.0214) and TMT labeled n-terminus (+229.1629 for TMT and +304.2071 for TMTpro) were applied as static modifications for all analyses. For proteome analysis, TMT or TMTpro on lysine was also set as a static modification. Methionine oxidation (+15.9949) and TMT or TMTpro on tyrosine were set as a dynamic modifications for all searches. For ubiquitylome searches, TMTpro on lysine and TMTpro-KGG on lysine (+418.2510) were considered as variable modifications. Peptide spectral matches for each run were filtered using line discriminant analysis (LDA) to a false discovery rate (FDR) of 2% and subsequently as an aggregate to a protein level FDR of 2%.

Quantification and statistical testing of TMT proteomics data was performed using MSstats[37]. Prior to MSstats analysis, peptide spectral matches (PSMs) were filtered to remove matches from decoy proteins; peptides with length <7; with isolation specificity <50%; with reporter ion intensity <256; and with summed reporter ion intensity (across all channels) <30,000. In addition, to separate peptides shared between SMARCA2 and SMARCA4, peptides matching to either protein were labeled as SMARCA2, SMARCA4, or SMARCA2/4 prior to MSstats analysis. This enables quantification of shared peptides to be performed as if they were a separate protein group. In the case of redundant PSMs (i.e., multiple PSMs in one MS run that map to the same peptide), PSMs were summarized by the maximum reporter ion intensity per peptide and channel and median equalized. In the case of redundant PSMs across fractions (i.e., redundant matching PSMs being found in multiple fractionated runs), PSMs were summarized by selecting the fraction with the maximum reporter ion intensity for each PSM. Protein level summarization was performed using a Tukey median polish approach. Differential abundance analyses between conditions were performed in MSstats based on a linear mixed-effects model per protein, and resulting two-sided p-values adjusted for multiple hypothesis testing by using the Benjamini-Hochberg procedure. For ubiquitylome analysis, the log2 ratio values of each ubiquitinated peptide were normalized to the corresponding protein measurement before visualization; if a protein log2 ratio was not measure the ubiquitylation measurement was carried forward unchanged.

## Immunofluorescence

Cells were plated in 384 well plates (PhenoPlate™ 384-well microplates, PerkinElmer) at 4000 cells/well overnight. Cells were subsequently treated for 24 h with a dose response of A947 prior to fixation with 4% formaldehyde for 15 min. Plates were washed three time with PBS, incubated with a blocking solution (10%FCS, 1%BSA, 0.1%Triton, 0.01% Azide, X-100 in PBS) for 1.5 h, and subsequently treated with primary antibody diluted 1:1200 in blocking buffer overnight at 4 °C. Following washing (3×) in PBS, cells were incubated with secondary antibodies (rabbit-Alexa 488, ThermoFisher A21206, 1:1000) for 1 h at room temperature in the dark. Hoechst H3570 (ThermoFisher H3570) at 1:5000 was added to the wells and the plates were incubated for an additional 30 min. Plates were wash 3× in PBS and imaged on an Opera Phenix™ High Content Screening System (PerkinElmer). Using Hoechst H3570 nuclear staining as a mask, nuclear SMARCA2 and SMARCA4 mean signal intensity were quantified.

## Cell viability assays

Cells were plated in 96-well plates at 500 cells per well and treated with a dose range of A947 starting with a highest concentration of 500 nM. After incubating for 7 days, viability was measured using CellTiter-Glo (Promega) reagent. Reagent was added directly to the cells at a 1/1 ratio of reagent to cell culture medium. Following a 15 min incubation, luminescence was measured using the multimode plate reader EnVision 2105 (PerkinElmer). Viability was normalized to DMSO treated control cells. Cell viability experiments were performed in triplicate cultures.

## Clonogenic assay

Cells (1500–5000, depending on doubling time) were plated in 12-well plates for 24 h prior to treatment with fresh media containing compounds at indicated concentrations. Fresh media containing compound was replaced every 3–4 days until cells reached confluence to stop culture. Colonies were visualized by staining with 0.5% crystal violet for 20 min at room temperature.

## Cell cycle analysis

Cell lines were treated for 48 h with a dose response of A947 prior to pulsing for 15 min with 10uM EdU. Cells were subsequently fixed in 4% paraformaldehyde for 10 min, washed 3 times with PBS and then blocked and permeabilized in PBS containing 10% FBS, 1% BSA, 0.1% TX-100, and 0.01% NaN3 for 1 h at room temperature. Permeabilization buffer was removed and the cells were washed 3× with PBS. The Click-iT® reaction was perform according to the manufacturer's (Invitrogen C10337) protocol. Following a 30 min incubation in the dark, the cells were washed 3 times with PBS. For nuclear staining, cells were treated with Hoechst 33342 (ThermoFisher) at 1:10000 for 10 min at room temperature. Cells were then washed again 3 times with PBS and imaged on Opera Phenix Plus High-Content Screening System (PerkinElmer). Image analysis was conducted with MATLAB standard and custom-written scripts (https://github.com/scappell/Cell_tracking). For each cell, the integrated nuclear Hoechst signal and EdU positivity were used to determine cell cycle phase and values were averaged for each treatment.

## Apoptosis

Live cell imaging for activated caspase-3/7 was performed using the IncuCyte® ZOOM (Essen Bioscience). Cells were seeded in 96 well plates and treated the next day with 100 nM A947 and/or 1uM AMG-176 in the presence of Caspase-3/7 Green Detection Reagent (Essen Bioscience). Fluorescence was monitored over a 48 h period, with data collection every 4 h. Five planes of view were collected per well using 10× objective. Both phase contrast and green channel were collected for all wells. Data are presented as fluorescent events per well.

## RNA sequencing and analysis

For in vitro gene expression studies, HCC2302 or HCC515 cells were treated with DMSO or A947(100 nM) for both 24 h and 96 h prior to isolation of total RNA using the MagMax mirVana total RNA isolation kit (ThermoFisher Scientific, A27828). In addition, HCC2302-shNTC and HCC2302-shSMARCA2 cells were treated for 168 h prior to isolation of RNA. For in vivo gene expression studies, total RNA was isolated from xenograft tissues as above. RNA concentrations were measured by NanoDrop8000 (ThermoFisher). Integrity of RNA was assessed by Bioanalyzer 2100 prior to library generation using 500 ng RNA. Libraries were prepared using the TruSeq Stranaded Total RNA Library Prep Kit (Illumina), multiplexed and sequenced on Illumina HiSeq2500 (Illumina) to generate ~30 M single end, 50 base pair reads. Raw sequencing reads for in vitro samples were mapped to the UCSC human genome (GRCh38/hg38) using GSNAP software[38]. In order to remove potential mouse stromal contamination for in vivo xenograft samples, raw sequencing reads were stripped of reads showing complete alignment to the UCSC mouse genome (mm10) using Xenome software[39]. Remaining reads not showing complete alignment to mm10 were then mapped to GRCh38/hg38 using GSNAP software. Gene expression counts were obtained by quantifying the number of reads uniquely mapping to each gene locus. Lowly expressed genes were removed from all samples using a high-pass filter for genes with at least 15 counts in at least 10% of samples (6 of 54). Quantile normalized $Log_2$ counts per million (LogCPM) of sufficiently covered genes were generated using the voom function of the limma analysis pipeline[40]. Differential gene expression analysis was performed using edgeR[41]. Significantly downregulated or upregulated genes were defined by a log fold change ($Log_2FC$) absolute value >1 and a false discovery rate (FDR) < 0.05. To evaluate in vivo samples, a consensus, putative SMARCA2 target gene set was defined by genes significantly downregulated by A947 treatment in both HCC515 and HCC2302 in vitro samples at both early (24 h) and late (96 h) time points. RNAseq data has been deposited in the Gene Expression Omnibus database under the accession code GSE205542.

## Mice

Female Crl:NU-Foxn1nu (NU/NU Nude) or CB17/Icr-$Prkdc^{scid}$/IcrIcoCrl (Fox Chase CB17) mice aged 6–8 weeks were purchased for Charles River laboratories. Mice were housed in individually ventilated cages within animal rooms maintained on a 14:10 h, light:dark cycle. Animal rooms were temperature and humidity-controlled, between 68–79 °F and 30–70% respectively, with 10–15 room air exchanges per hour. Mice received food and water ad libitum and were allowed to acclimate for 1–2 weeks before being used for experiments. All animal work was approved and conducted in accordance with the approval from the Institutional Animal Care and Use Committee (IACUC). All animal studies complied with the ethical regulations and humane endpoint criteria according to the NIH Guidelines for the Care and Use of Laboratory Animals. Genentech is an AAALAC-accredited facility and all animal activities in the research studies were conducted under protocols approved by the Genentech Institutional Animal Care and Use Committee (IACUC). The maximal subcutaneous tumor size/burden allowed (2000 mm³) was not exceeded in this study. Euthanasia was carried out by anesthetization through isoflurane inhalation followed by cardiac exsanguination.

## Xenograft seeding

HCC515 and HCC2302 were propagated in RPMI-1640/10%FBS (Gibco, #61870-036), and Calu-6 in EMEM/10%FBS (ATCC, #30-2003). The cells were lifted with Trypsin (0.25%) (Gibco, #25200-056), spun down at 500 × g, washed at least 3 times with DPBS (Gibco, #14190-144), and the pellet resuspended in 50%Matrigel (Corning, #354234)/50% phenol-red free RPMI-1640 (Gibco, #11835-030). Implant cell numbers for HCC515 and Calu-6 was 5 × 10^6 and for HCC2032 was 10 × 10^6 in a 100 or 200 ul volume.

## HCC2302, Calu-6, HCC515 xenograft mouse models

Tumor cells were implanted into the right flank of the NU/NU or FOX CHASE SCID mice. The Tumor growth was monitored daily, and tumors were measured twice a week using digital calipers. Tumor volume was determined using the following formula (*width × width × length)/2), where all measurements are in mm and the tumor volume is in mm³*. The treatment started once the average tumor volume reached 150–200 mm³. Treatment was started ~3 weeks after cell implantation. The animals were randomly assigned into separate groups ($n = 6$–10 animals per group) such that each group had nearly equal starting average tumor volume. Treatment groups were randomly assigned into groups treated with vehicle and A947. A947 was dosed 5 mg per kg of body weight into the lateral tail vein intravenously once a week or every other week for tumor growth studies or only once for PK/PD studies. A947 was formulated for intravenous dosing in 10% Hydroxypropyl Beta Cyclodextrin (HP-b-CD) and 50 mM sodium acetate in water (pH 4.0). All dosing solutions were filtered prior to injection using a 0.2 micron filter to endure sterility. Mice were weighed twice a week, and dosing was performed the treatments were given according to the mouse's individual weight. Mice were euthanized using an IACUC approved method of euthanasia when an individual mouse reached a maximum tumor size humane endpoint, defined according to institutional policy concerning tumor endpoints in rodents. In addition, to prevent excessive pain or distress, the mice were euthanized if the tumors became ulcerated or if the mice showed any signs of ill health. Post euthanasia, blood and various tissues including tumors were collected for further analyses.

Analyses and comparisons of tumor growth were performed using a package of customized functions (https://github.com/wfforrest/maeve) in R (Version 3.4.2 and 3.6.2; R Foundation for Statistical Computing; Vienna, Austria), which integrates software from open source packages (e.g., lme4, mgcv, gamm4, multcomp, settings, and plyr) and several packages from tidyverse (e.g., magrittr, dplyr, tidyr, and ggplot2)[42]. Briefly, as tumors generally exhibit exponential growth, tumor volumes were subjected to natural log transformations before analysis. Estimates of group-level efficacy were obtained by calculating percent tumor growth inhibition (TGI). This value represents the percent difference between the area under the curves (AUCs) of the treatment and reference group fits which are calculated after back-transforming tumor volumes to the original scale, correcting for starting tumor burden, and averaging over a common time period. Positive values indicate anti-tumor effects, with 100% denoting stasis and values >100% denoting regression (negative values indicate a pro-tumor effect).

## Xenograft tissue processing for pharmacodynamic assessments

Xenografts were harvested, divided into pieces, flash frozen and stored at −80C. For transcript-based pharmacodynamic readouts, RNA was isolated using the MagMAX mirVana total RNA isolation kit. For protein-based pharmacodynamic readouts, RIPA + Halt protease inhibitor (Thermo Fisher, #74830) was used at 400 ul per tube, regardless of tumor weight. A steel ball was used in each sample in the TissueLyzer at 26 Hz for 4 min. The homogenization block was stopped half way through the process and the block flipped over for the duration of the process. Lysates were sonicated for 30 s on 20 Hz in an icy bath. The lysates were spun clean at 15,000 RPM for 15 min at 4 C. Samples were then assessed for concentration by BCA at a dilution of 1:25. Western samples were prepared at 1 ug/ul in SDS-PAGE loading buffer/denaturing agent (Life Technologies, #NP0007 + #B0009) and denatured at 95 C for 5 min. Samples were used immediately or frozen at −20C until blotted.

## Immunoblotting (in vivo samples)

Protein (8 ug) was loaded on 4–15% Criterion Tris/Glycine gels (Bio-Rad, #5671085) and run for 60 min at 150 constant volts in 1X Tris/Glycine buffer (Bio-Rad, #1610732). Protein was transferred from gels to nitrocellulose with Bio-Rad Turbo on default setting. All blots were air-dried, rehydrated with TBS and blocked for 1 h at RT in 5% BSA in TBST (0.1%). Blots were exposed to primary antibody in 5% BSA in TBST (0.1%) overnight at 4 C. Blots were washed 3× with TBST (0.1%), 5 min per wash, at RT. Secondary antibody was added at 1:18,000: anti-rabbit-HRP (CST 7074) and/or anti-mouse-HRP (CST 7076) in 5% BSA in TBST (0.1%). Blots were incubated at RT for 1 h. Blots were washed 3 times in TBST (0.1%) for 5 min each wash at RT. All incubations and washing were done while rocking. Signal was developed with 6 ml of Femto Max ECL substrate (ThermoFisher, #34094) for 4 min and blots read on ChemiDoc. Densitometry was performed with ImageLab.

## Quantitative RT-PCR

Gene expression levels were determined by Taqman using the following Taqman gene expression assays (*KRT80*, Hs01372363_g1; *PLAU*, Hs01547051_g1; GUSB, Hs00939627_m1) and the Taqman RNA-to-Ct 1-Step kit (ThermoFisher Scientific). Analysis is performed using QuantStudio™ 7 Flex Real-Time PCR System (ThermoFisher Scientific). Expression levels are normalized ($2^{-\Delta Ct}$) to the housekeeping gene, *GUSB* and presented relative to expression levels in vehicle-treated tumors.

## Immunohistochemistry

Immunohistochemistry was performed on formalin-fixed, paraffin embedded 5 μm thick sections using a DAKO autostainer (Agilent, Santa Clara, CA) and target antigen retrieval (Agilent). A polyclonal rabbit antibody against SMARCA2 (Sigma, Cat# HPA029981) and a rabbit monoclonal antibody against SMARCA4 (AbCam, Clone EPN-CIR111A) were used as primary antibodies at a final concentration of 0.5 and 0.11 ug/ml, respectively. Secondary antibodies were a biotinylated goat anti-rabbit antibody (SMARCA4) or a biotinylated donkey anti-rabbit antibody (SMARCA2) and specifically bound antibody was detected using diaminobenzidine and an avidin-biotin-based peroxidase reaction (ABC-Peroxidase Elite, PK-6100, Vector Laboratories). Tissue sections were counterstained with Mayer's hematoxylin.

## Pharmacokinetics assessment

For Tumor: After the addition of 30 μL of acetonitrile per 10 mg of tumor, tumor samples were homogenized. 100 μL of tumor homogenate was pipetted out for analysis. After the addition of 50 μL of DMSO:acetonitrile 1:1 (v/v), 20 μL of 2 μg/mL propranolol in methanol:water 1:1 (v/v) as internal standard (IS), and 200 μL of chilled acetonitrile, tumor samples were vortexed and centrifuged at 3500 *rpm* for 30 min. 2 μL of supernatant was injected onto an AB Sciex API 4000 LC-MS/MS system coupled with a Shimadzu Prominence HPLC for analysis. For plasma: 20 μL of DMSO:acetonitrile 1:1 (v/v) and 20 μL of 2 μg/mL propranolol in methanol:water 1:1 (v/v) as internal standard is added into 20 μL of plasma sample, then 200 μL of chilled acetonitrile was added to precipitate protein. Samples were vortexed and centrifuged at 3500 *rpm* for 10 min. 2 μL of supernatant was injected onto an AB Sciex API 4000 LC-MS/MS system coupled with a Shimadzu Prominence HPLC for analysis. LC separation was performed on a Phenomenex Synergi Polar-RP column (4 μm, 80 Å, 2 × 50 mm) with 0.1% acetic acid 1 mM ammonium acetate in water as mobile phase A and 50 mM acetic acid in acetonitrile as mobile phase B. A gradient elution at 0. 5 mL/min started with 30% B. B component was increased linearly to 75% in 0.5 min. After holding at 75% B for 1.5 min, the column was reequilibrated with 30% B for 0.75 min. Mass spectrometric detection was performed with TurboSpray ionization in positive ion mode.

## Drug combination screens

A chemical library comprising 723 compounds arrayed in nine-point dose–response was screened in the absence or presence of a fixed dose of 100 nM A947. Compounds were obtained from in-house synthesis or purchased from commercial vendors. Cells were dispensed using the Multidrop™ Combi Reagent Dispenser (Thermo Scientific; Waltham, MA) into 384-well, black, clear-bottom plates (Corning, Tewksbury, MA) at seeding densities previous determined to achieve ~70–80% confluence at the final time point of the assay. Following overnight culture, compounds were dispensed using the Bravo Automated Liquid-Handling Platform (Agilent; Santa Clara, CA). Following a 5 day culture period, 25 μL CellTiter-Glo® reagent was added using a MultiFlo™ Microplate Dispenser (BioTek). Cell lysis was induced by mixing for 30 min on an orbital shaker prior to incubating plates at room temperature for 10 min to stabilize the luminescent signal. Luminescence was read by a 2104 EnVision® Multilabel Plate Reader (PerkinElmer; Waltham, MA). Data was processed using Genedata Screener®, Version 15 (Genedata; Basel, Switzerland), with a four-parameter Hill equation using compound dose–response data normalized to the median of 42 vehicle-treated wells on each plate. A "Robust Fit" strategy was also employed by Genedata Screener®, which is based on Tukey's biweight and is resistant to outlier data. The reported absolute $IC_{50}$ is the dose at which cross-run estimated inhibition is 50% relative to DMSO control wells. Data are plotted as the difference in the $IC_{50}$ in the presence versus absence of A947. For matrix-based combinations, cells were seeded and assessed for viability in the same manner as described for the chemical library screen. Cells were treated with A947 (top concentration, 5 uM) in combination with AMG-176 (top concentration, 500 nM) or S63845 (top concentration, 500 nM) in a threefold dilution, 9 × 9 matrix. Drug synergy was assessed using the Bliss independence model[43] and data is presented as excess matrix heatmaps that represent differences between the observed and predicted values determined from the Bliss model for each concentration pair.

## Data availability

Raw RNAseq data generated in this study has been deposited in the Gene Expression Omnibus database under the accession code GSE205542. The mass spectrometry proteomics data are deposited with the ProteomeXchange Consortium via the PRIDE partner repository with the dataset identifier PXD036865. Source data are provided with this paper. The remaining data are available within the paper, Supplementary Information or Source Data File. Source data are provided with this paper.

## Code availability

The script used to analyze cell cycle images can be found in https://github.com/scappell/Cell_tracking.

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

## Acknowledgements

We would like to thank the following individuals for their assistance with reagent generation, experimentation and/or helpful discussions: Keith Anderson, Yvonne Kschonsak, Linda Rangell, Chad Liu, Mark Bookbinder, Greg Cadelina, Kim Davenport, Dean DiNicola, and Debbie Gordon.

## Author contributions

J.C. and R.L.Y. designed, analyzed and interpreted experiments and wrote the paper, X.Y., E.R., T.J., T.K.C., T.H., L.S., C.Q., A.H., E.B., E.L., D.D., H.K., S.F.Y., G.D.R., E.C., M.G., B.D.H., X.C., and R.H. designed, performed and analyzed experiments, C.M.R., S.M., M.M, F.B., J.W., J.P., P.S.D., M.B. designed, analyzed and interpreted experiments. All authors commented on the paper.

## Competing interests

Authors are employees of Genentech and/or Arvinas (as defined in affiliations) and own shares of Roche or Arvinas stock, respectively. The authors declare no competing interests.
