## [Peer Review File · Nature Communications]

Reviewers' Comments:

Reviewer #1:

Remarks to the Author:

In the manuscript the authors report a new moderately specific SMARCA2 degrader molecule and show this has efficacy in vitro and in vivo on SMARCA4 mutant cells. Several non-selective inhibitors of the highly homologous SMARCA2/4 proteins exist, but this paper shows the designed PROTAC strategy yields some specificity for SMARCA2. The mechanism that controls this specificity is relatively unexplored, but nevertheless the paper describes a useful molecule that will help investigation of SMARCA4 mutant cancers. Overall, the experiments are well-described, cautiously interpreted, and the conclusions support the main point of the paper.

I have only minor comments on experimental presentation and request for clarifications. Notably, I am not a chemical synthesis expert and have not evaluated if the information on synthesis of the PROTAC is sufficient.

1. In figure 3C, I would encourage inclusion of the control cell lines in the main figure. It was difficult to compare G1 cells without seeing what the controls look like.
2. Figure 4a, and extended data. These plots were difficult to read with the multiple axis. I'd suggest breaking this into independent figures.
3. In several places in the discussion there are results presented as data not shown. These statements should be removed or the data should be presented. Otherwise, future work cannot accurately build on the results.
4. I did not see an accession number for the RNAseq data presented in Figure 3d

Reviewer #2:

Remarks to the Author:

The authors have generated a proteolysis targeting chimera (PROTAC) degrader using a molecule that binds the bromodomains of SMARCA2, SMARCA4 and PBRM1 to the E3 ubiquitin ligase VHL. This molecule (A947) seems to exhibit a potent and, to some extent, selective degradation of SMARCA2. There has been at least another previous publication on the generation of a different PROTAC targeting SMARCA2 for degradation, which is referred to by the authors in their manuscript.

Therapies targeting SMARCA4-mutant tumors are of great interest, given the relatively high frequency of these alterations in many solid tumors, including lung cancer, among other highly aggressive cancers. Therefore, the work is valuable in terms of clinical purposes.

Comments:

- 1.- The authors use a panel of lung cancer cell lines that are mutant for SMARCA4 and a control panel of wild-type SMARCA4 cells. It is known that, for still unknown reasons, some SMARCA4-mutant cells show very low or complete absence of SMARCA2 protein, by western-blot. They provide the information on the genetic alterations in SMARCA4 in these cells (extended Table 3) and on the mRNA levels (RNA-seq of the databases) of SMARCA4 and SMARCA2. However, the authors should not rely on the mRNA to determine the presence or absence of SMARCA2 in each of the cell lines. Instead, they should perform a western-blot to determine the total protein levels of SMARCA4 and SMARCA2 of every one of the cell lines included in the study. The results should be included in a main figure.
- 2.- To provide more robust data, the authors need to show the reduction in SMARCA2 protein levels with A947 administration and determine if this reduction parallels effects on cell growth inhibition in a wide panel of lung cancer cell lines (among those used to determine the effects of the drug on the growth inhibition cell), not just a single cell line as shown in the present work. For this, it would be appropriate to carry out a western blot of the SMARCA2, SMARCA4 (for SMARCA4-wt cells) and PBRM1 proteins after the administration of the drug.

- 3.- Some immunostainings of SMARCA2 and SMARCA4 (or at least of SMARCA2) should be performed in the animal models to more clearly evidenced the reduction in the levels of these proteins, as compared to the controls. It is intriguing that, in their work, the authors detect SMARCA4 protein by wb in most of the SMARCA4-mutant cells, since most truncated SMARCA4 proteins are usually degraded by the cells (non-sense mediated decay mechanism).
- 4.- Statistics are lacking in the figure 3, fig 5c, and in ext. data fig 6.
- 5.- Is the data in figure 4g from a WB? Please indicate it and show the blot. In fig 5b, shouldn't some SMARCA4wt cells been included for comparison?
- 6.- There are only 5 main figures but 10 extended data figures. The authors could move some of the figures from the extended data, into main figures. Extended data 1 and 5a could be added to figure 1. Extended data fig 6b and c could be moved to figure 3. Extended fig. 10 moved to Fig 4...
- 7.-To make the reading easier, please include the name of the cell line used in each of the figures.

Reviewer #3:

Remarks to the Author:

This is an excellent degrader development story detailing the discovery and detailed characterization of a heterobivalent degrader of SMARCA2 with selectivity versus SMARCA4. The conclusions are well supported by the data provided and detailed control studies are performed. A detailed experimental section is included and the chemistry is well described. The authors also do an excellent job characterizing the compound using in vivo tumor models which only a few degrader papers to-date have done. The paper is also nicely written, easy to read and the conclusion section is good. The figures are nice and easy to follow. I think the paper deserves to be published without further modification.

Specific Review Questions:

- What are the noteworthy results?

Discovery of SMARCA2 selective degraders, corroboration of selective cytotoxicity in cells harboring loss of function in SMARCA4, demonstration of in vivo efficacy.

- Will the work be of significance to the field and related fields? How does it compare to the established literature? If the work is not original, please provide relevant references.

Helps advance the field of small molecule degraders, establishing differentiation relative to inhibitors, demonstrates ability to get paralog selectivity.

- Does the work support the conclusions and claims, or is additional evidence needed?

Conclusions are well supported by the provided data.

- Are there any flaws in the data analysis, interpretation and conclusions? Do these prohibit publication or require revision?

None noted.

- Is the methodology sound? Does the work meet the expected standards in your field?

Yes

- Is there enough detail provided in the methods for the work to be reproduced?

Yes

For ease in reviewing our changes to the manuscript, we additionally uploaded a marked-up word version of the main body and methods section of the manuscript to more readily allow the reviewers to see the exact changes we made. Figures are embedded within the final manuscript. We look forward to your review of this revised manuscript.

RESPONSE TO REVIEWER COMMENTS

Reviewer #1 (Remarks to the Author):

In the manuscript the authors report a new moderately specific SMARCA2 degrader molecule and show this has efficacy in vitro and in vivo on SMARCA4 mutant cells. Several non-selective inhibitors of the highly homologous SMARCA2/4 proteins exist, but this paper shows the designed PROTAC strategy yields some specificity for SMARCA2. The mechanism that controls this specificity is relatively unexplored, but nevertheless the paper describes a useful molecule that will help investigation of SMARCA4 mutant cancers. Overall, the experiments are well-described, cautiously interpreted, and the conclusions support the main point of the paper.

We thank the reviewer for their commentary.

I have only minor comments on experimental presentation and request for clarifications. Notably, I am not a chemical synthesis expert and have not evaluated if the information on synthesis of the PROTAC is sufficient.

1. In figure 3C, I would encourage inclusion of the control cell lines in the main figure. It was difficult to compare G1 cells without seeing what the controls look like.

We have added the control lines to the main figure (revised Fig. 3D)

2. Figure 4a, and extended data. These plots were difficult to read with the multiple axis. I'd suggest breaking this into independent figures.

We have separated out the PK data from the tumor PD data. (Fig 4a and revised Extended Data Figure 5a)

3. In several places in the discussion there are results presented as data not shown. These statements should be removed or the data should be presented. Otherwise, future work cannot accurately build on the results.

We have made the suggested edits, removing any statements referencing data not shown.

4. I did not see an accession number for the RNAseq data presented in Figure 3d

Our RNAseq data has been accessed under GEO205542. We have updated the methods accordingly.

Reviewer #2 (Remarks to the Author):

The authors have generated a proteolysis targeting chimera (PROTAC) degrader using a molecule that binds the bromodomains of SMARCA2, SMARCA4 and PBRM1 to the E3 ubiquitin ligase VHL. This molecule (A947) seems to exhibit a potent and, to some extent, selective degradation of SMARCA2. There has been at least another previous publication on the generation of a different PROTAC targeting SMARCA2 for degradation, which is referred to by the authors in their manuscript.

Therapies targeting SMARCA4-mutant tumors are of great interest, given the relatively high frequency of these alterations in many solid tumors, including lung cancer, among other highly aggressive cancers. Therefore, the work is valuable in terms of clinical purposes.

We thank the reviewer for their commentary.

Comments:

1.- The authors use a panel of lung cancer cell lines that are mutant for SMARCA4 and a control panel of wild-type SMARCA4 cells. It is known that, for still unknown reasons, some SMARCA4-mutant cells show very low or complete absence of SMARCA2 protein, by western-blot. They provide the information on the genetic alterations in SMARCA4 in these cells (extended Table 3) and on the mRNA levels (RNA-seq of the databases) of SMARCA4 and SMARCA2. However, the authors should not rely on the mRNA to determine the presence or absence of SMARCA2 in each of the cell lines. Instead, they should perform a western-blot to determine the total protein levels of SMARCA4 and SMARCA2 of every one of the cell lines included in the study. The results should be included in a main figure.

We have incorporated a new figure (revised Fig. 3b) evaluating SMARCA2, SMARCA4, PBRM1 and a loading control across the full panel of cell line models utilized in the manuscript. Given the number of cell lines, we were unable to run all cell lines in one immunoblot. Hence, we separated into 2 immunoblots and purposely included models representative of both SMARCA4 mutant and wild-type into each blot; in order to provide the reader better perspective on protein levels between these states.

2.- To provide more robust data, the authors need to show the reduction in SMARCA2 protein levels with A947 administration and determine if this reduction parallels effects on cell growth inhibition in a wide panel of lung cancer cell lines (among those used to determine the effects of the drug on the growth inhibition cell), not just a single cell line as shown in the present work. For this, it would be appropriate to carry out a western blot of the SMARCA2, SMARCA4 (for SMARCA4-wt cells) and PBRM1 proteins after the administration of the drug.

This is an important point that we apologize that we missed in the initial submission. We addressed the reviewers concern 2 different ways.

- (1) We evaluated a dose response of A947 across a panel of 7 SMARCA4 mutant and 7 SMARCA4 wild-type models by immunofluorescence and monitored degradation of SMARCA2 and SMARCA4. The calculated DC_{50} and 95% confidence interval values are incorporated into Supplemental Table 3 (columns L/M and N/O). Although not included in the manuscript, we have pasted the dose response curves from which these DC_{50} 's were calculated at the end of this letter for full transparency. Despite the differences in cellular growth inhibition between SMARCA4 mutant and WT cell line models, this was not solely due to differences in the ability of A947 to degrade SMARCA2 (Extended Data Fig. 4b). A947 was equally potent on SMARCA2 between these models. Also, although there was a range of IC_{50} 's for growth inhibition within SMARCA4 mutant and WT models, we did not observe a direct correlation with growth inhibition (IC_{50}) and degradation of SMARCA2 within the SMARCA4 mutant or within the SMARCA4 WT models. The differences in growth inhibition within the SMARCA4 WT models could also not be accounted for due to differences in SMARCA4 degradation (Extended Data Fig. 4d). The plots to highlight a lack of correlation are incorporated as Extended Data Figures 4b and d and we additionally modified the text to communicate this point.
- (2) We additionally evaluated a dose response of A947 across a smaller panel of 4 SMARCA4 mutant and 4 SMARCA4 wild-type models and monitored not only degradation of SMARCA2 and SMARCA4, but PBRM1 by Western blotting given that we did not have a robust immunofluorescence assay. We selected cell lines that represented the largest differences in growth inhibition (i.e. the IC_{50} outliers). This data is incorporated in Extended Data Figure 4c. Differences in growth inhibition could also not be accounted for by differences in degradation of PBRM1 between or within SMARCA4^{mut} or SMARCA4^{WT} models.

3.- Some immunostainings of SMARCA2 and SMARCA4 (or at least of SMARCA2) should be performed in the animal models to more clearly evidenced the reduction in the levels of these proteins, as compared to the controls. It is intriguing that, in their work, the authors detect SMARCA4 protein by wb in most of the SMARCA4-mutant cells, since most truncated SMARCA4 proteins are usually degraded by the cells (non-sense mediated decay mechanism).

We have evaluated SMARCA2 levels by immunohistochemistry to provide orthogonal evidence demonstrating the reduction in SMARCA2 *in vivo* (revised Extended Data Fig. 5b). Regarding the commentary on SMARCA4, we apologize if this was misunderstood. Indeed, SMARCA4 transcript undergoes non-sense mediated decay and we observe little evidence for strong SMARCA4 protein levels in cell lines with SMARCA4 truncating mutations (see revised Figure 3b) (note: cell lines with SMARCA4

missense mutations do still express SMARCA4 protein) However, we do observe a Smarca4 band by Western blotting from the *in vivo* xenografts of human SMARCA4 mutant cell lines due to antibody cross-reactivity with murine Smarca4 (Extended Data Figure 8). This is further highlighted by the immunohistochemistry evaluating SMARCA4 levels in xenografts (Extended Data Fig. 5b), whereby a lack of signal is observed within the tumor epithelium, however expressed within the stromal infiltrate.

4.- Statistics are lacking in the figure 3, fig 5c, and in ext. data fig 6.

We have added the respective statistics.

5.- Is the data in figure 4g from a WB? Please indicate it and show the blot. In fig 5b, shouldn't some SMARCA4wt cells been included for comparison?

Yes, the data from Figure 4g is derived from Western blot. We modified the legend to indicate this and included the Western blot data in revised Extended Data Fig. 8a to provide direct comparison with the SMARCA4^{mut} models in Extended Data Fig. 8b and 8c.

We have included 2 SMARCA4^{WT} models in Extended Data Fig. 9 to demonstrate a lack of synergy between SMARCA2 degradation and MCL1 inhibition.

6.- There are only 5 main figures but 10 extended data figures. The authors could move some of the figures from the extended data, into main figures. Extended data 1 and 5a could be added to figure 1. Extended data fig 6b and c could be moved to figure 3. Extended fig. 10 moved to Fig 4...

We have moved Extended Data 1 to Main Figure 1d.

We have moved Extended Data 5a to Main Figure 1g.

Extended Data Figure 6c was moved to Main Figure 3d

Due to size constraints for the revised Figures 3 and 4, we were unable to move Extended Data Figures 6b and 10, respectively. However, we did re-arrange the Extended Data Figures and combine a few to make the extended data easier for the reader to visualize. As such, we have shortened the Extended Data figures from 11 to 9. A summary of the changes made to the original Extended Data Figures are:

Ext Fig 1.

- Moved to Main body Fig. 1d

Ext Fig 2.

- Not modified, revised Extended Data Fig. 1

Ext Fig 3.

- Not modified, revised Extended Data Fig. 2

Ext Fig 4.

- Not modified, but combined with Ext Data Fig. 4d to make revised Extended Data Fig. 3

Ext Fig 5.

- Moved 5a to Main body Fig. 1g
- Combined 5b with Ext Data Fig. 4d to make revised Extended Data Fig. 3

Ext Fig 6.

- Moved 6c to Main body Fig. 3d
- 6a, 6b, revised Extended Data Fig. 4a, 4e, respectively

Ext. Fig 7.

- Split into 2 separate graphs as per Reviewer #1, revised Extended Data Fig. 5a

Ext. Fig 8-11.

- Not modified, revised Extended Data Figures 6-9.

7.-To make the reading easier, please include the name of the cell line used in each of the figures.

We have added cell line names to the figures.

Reviewer #3 (Remarks to the Author):

This is an excellent degrader development story detailing the discovery and detailed characterization of a heterobivalent degrader of SMARCA2 with selectivity versus SMARCA4. The conclusions are well supported by the data provided and detailed control studies are performed. A detailed experimental section is included and the chemistry is well described. The authors also do an excellent job characterizing the compound using in vivo tumor models which only a few degrader papers to-date have done. The paper is also nicely written, easy to read and the conclusion section is good. The figures are nice and easy to follow. I think the paper deserves to be published without further modification.

We thank the reviewer for their commentary.

Reviewers' Comments:

Reviewer #1:

Remarks to the Author:

The authors have addressed all my concerns. I believe this should be published.

Reviewer #2:

Remarks to the Author:

The authors have adequately addressed all the comments and concerns raised.